# Experimental Evidence for Double Quaternary Azeotropy’s Existence

**DOI:** 10.3390/e25070980

**Published:** 2023-06-26

**Authors:** Anastasia Frolkova, Valeriy Zhuchkov, Alla Frolkova

**Affiliations:** Lomonosov Institute of Fine Chemical Technologies, MIREA—Russian Technological University, 119454 Moscow, Russia; frolkova_nastya@mail.ru (A.F.); frolkova@gmail.com (A.F.)

**Keywords:** double azeotropy, phase diagram, vapor–liquid equilibrium, quaternary system, singular points, azeotropy rule

## Abstract

The phase equilibrium in an acetonitrile + water + cyclohexene + chloroform system was studied at 101.3 kPa. A prediction regarding the internal structure of the composition tetrahedron (presence/absence of one/two internal singular points) was made using thermodynamic modeling in AspenPlus V.10.0. The existence of two internal quaternary azeotropes (of node type with a minimum boiling point and of saddle type) was confirmed as a result of a full-scale experiment. Thermodynamic-topological analysis of the structure of phase equilibrium diagrams was carried out to confirm the correctness of the diagram construction.

## 1. Introduction

Mathematical modeling is the most accessible and fastest method of studying the properties of components and their mixtures. An important constituent of the computational experiment is to check the adequacy of the model used (sets of parameters of the selected equations). The latter is verified by comparing experimental and calculated data on the properties of the components, as well as the mixtures they form. Unfortunately, the number of experimental data on phase equilibrium (vapor–liquid, liquid–liquid, etc.), data on azeotropy sharply decreases with an increase in the number of components. An adequate description of the phase equilibrium of the constituents of a multicomponent mixture does not always guarantee the reproduction of the features of the internal structure of the composition simplex [1]. The literature contains a number of works devoted to the problems of phase equilibrium modeling using different models [2,3,4,5,6,7,8,9]. Recommendations and restrictions on the use of certain models are formulated. The analysis of the works shows that there is no universal method (model), and in some cases (especially when studying systems with complex phase behavior) it is necessary to consider and compare several models. However, it is not worth limiting ourselves to thermodynamic modeling when studying the phase equilibrium of multicomponent systems. It is necessary to involve both qualitative methods and competently plan a full-scale experiment. 

As a rule, qualitative methods of diagram research are aimed at analyzing and predicting the structure. For example, several structures of the internal space may correspond to the same structure of the boundary space of the phase diagram [10,11,12]. In order to reveal the true structure of the diagram of an n-component system, it is necessary to determine the types of boundary singular points with a number of components (n − 1) relative to the complete phase diagram: these are binary azeotropes for ternary systems, ternary azeotropes for quaternary systems, etc. If the types and Poincaré indexes of all boundary singular points in the composition simplex are known, the presence/absence of an internal azeotrope can be determined by solving the azeotropy rule [13,14]. It is important to note that the solution of the rule, provided there is no internal azeotrope or the presence of two internal (biazeotropy) azeotropes, will be identical.

The use of the above techniques in the complex allows competently planning a full-scale experiment, minimizing the amount of work to find the internal structure of the phase diagram.

This article is devoted to the study of phase equilibrium in the acetonitrile + cyclohexene + water + chloroform system. The first three components are constituents of a multicomponent mixture for cyclohexanone production [15,16], and chloroform is considered as a potential extractant at one of the separation stages [17], since extraction is referred to as an energy-efficient separation technique [18]. 

This system is distinguished by a complex phase behavior: the presence of azeotropes with different numbers of components, in two and three liquid phases [19,20,21,22]. Assumptions about the possible structures of the composition tetrahedron are made, the existence of two quaternary azeotropes is predicted, and a plan for conducting a full-scale experiment is drawn up using computational experiment and qualitative methods (thermodynamic-topological analysis).

## 2. Materials and Methods

Since the parameters of the binary interaction of the NRTL equation [23], which allow adequately describing the phase equilibrium in binary constituents, as well as the azeotropes’ characteristics (composition and boiling point) are known, thermodynamic modeling was used at the first stage to predict the internal structure of the phase diagram. At the second stage, a thermodynamic-topological analysis of the diagram was used in order to plan and correctly conduct a full-scale experiment. The final—the third stage—was an experimental study and confirmation of the structure.

### 2.1. Thermodynamic Modeling

The thermodynamic modeling was carried out using AspenPlus V.11.0 software and the non-random two-liquid (NRTL) model. The NRTL equation parameters were taken from the literature [1,19,20]. The adequacy of modeling was assessed by comparing experimental and calculated values of azeotropic characteristics (composition and boiling point). The relative standard uncertainty (u_r_) was calculated using the formula: (1)$$u_{r}=\frac{u}{a^{\exp}}=\frac{\left|a^{\exp}-a^{\mathrm{cal}}\right|}{a^{\exp}}$$

The results obtained (Table 1) indicate the correctness of the azeotropic properties’ reproduction (the relative standard uncertainty does not exceed 0.05).

UNIFAC, UNIF-LL and NRTL-HOC models were considered as alternative equations. The results of the description of the azeotropic properties by these models (Appendix A), as well as the description of the LLE equilibrium in binary constituents (Appendix A) and LLLE in ternary constituents (Appendix A) by all four models are given in the Appendix A. The experimental data on LLE and LLLE were taken from [19,20,26,27]. 

### 2.2. Thermodynamic-Topological Analysis

The equation of the azeotropy rule was used as the basis for the analysis of the VLE diagram structure. In this paper, the equation proposed by L.A. Serafimov [13,14] is considered:(2)$$2\left(N_{m}^{+}+S_{m}^{+}-N_{m}^{-}-S_{m}^{-}\right)+\sum_{0}^{m-1}\left(N_{i}^{+}+S_{i}^{+}-N_{i}^{-}-S_{i}^{-}\right)=E,$$
where *m* (*i*)—the dimension of the element on which the singular point (azeotrope, pure component) is located (*m* refers to the inner space, *i*—refers to the boundary space), $E=1+{\left(-1\right)}^{m}$—Euler characteristic. 

The type and Poincaré index of the singular points depends on the signs of the characteristic roots obtained as a result of solving a linearized system of nonlinear differential equations describing the distillation process. The procedure for determining the type and index of singular points is described in detail in [11,12,13,14]. 

The number of characteristic roots for quaternary systems is three. Points of the stable (unstable) node type will have an index equal to +1 (−1). The index of the saddle point depends on the number of negative roots (two negative roots, the index is +1; one negative is the index −1). Some points can be saddle nodes relative to the boundary space, i.e., have an index equal to zero. Such points are not included in Equation (2). Having determined the sum of the indices of all boundary singular points, it can be concluded whether there is an internal azeotrope in the system. If the sum is +2 (−2), the system has a quaternary azeotrope with the index −1 (+1). If the sum is zero, then two cases are possible: (a) there is no internal azeotrope in the system; (b) there are two internal azeotropes. In case (b), two internal azeotropes will always be characterized by indexes with different signs. In addition, if it is known that there is at least one internal azeotrope in the system, then the presence of the second one can also be determined by solving Equation (2). The main difficulty is the determination of the types of singular points corresponding to ternary azeotropes. To establish the latter, the authors of [11] recommend a study of the change in the boiling point of a quaternary mixture in the vicinity of the composition of a ternary azeotrope. To do this, it is advisable to use ebulliometric titration.

After determining the internal azeotrope index, it is necessary to establish its type. For this purpose, the structure of all separatric manifolds to which this singular point belongs should be analyzed. If a point is represented by a node (stable or unstable) relative to all surfaces, then the azeotrope is a node. If at least on one surface a point is represented by a saddle, then the azeotrope is a saddle.

### 2.3. Experimental Procedures

#### 2.3.1. Chemicals

The purity of the chemicals was verified chromatographically using a GLC with a flame ionization detector (CHROMOS −1000, Russia) and an FFAP capillary column (length 50 m, i.d. 0.32, film thickness 0.50, column temperature 373 K, and injector and detector temperatures 523 and 493 K). The chemical properties (purity, boiling temperature, density) and both experimental and literature data are given in Table 2. All chemicals were used without further purification. Water was bidistilled.

#### 2.3.2. Ebulliometric Titration

The choice of this method was primarily due to the device design feature, in particular related to the possibility of intensive mixing of the liquid (the system contained areas with two and three liquid phases). The ebulliometric titration was carried out in a modified Sventoslavskiy still [29] (Figure 1). The device was equipped with a magnetic stirrer (to prevent mixture splitting) and with a special internal heating element. The manostating system was used to maintain the necessary pressure (101.3 kPa ± 0.1 kPa). The VLE mixture was pumped by a Cottrel pump. The boiling point was fixed using an electronic laboratory thermometer, the LT-300 (Russia) (an uncertainty ± 0.05 K) when the equilibrium state was reached (the temperature did not change for 30 min; the number of drops in the drop counter remained 120 per minute) [30].

The task of ebulliometric titration was to add small amounts of chloroform to the ternary mixture of ACN + CHEN + W (to its total concentration of about 30% mass.) and determine the trend of boiling point changes. The experiments were carried out in parallel in two identical devices; the time for achieving equilibrium, during which the parameters did not change, was at least 30 min. The trend of the boiling point change was almost the same; the points fell on the same curve. The latter was also confirmed by mathematical modeling.

## 3. Results

A scan and the complete structure of the composition tetrahedron were constructed based on data from Table 1, Appendix A. The scan is characterized by the same structure for all models (Figure 2).

Different structures of the total composition space can correspond to the same structure of the tetrahedron scan: without internal singular points, or with one or two internal azeotropes. Examples of such structures are shown in Figure 3.

Unfortunately, the capabilities of the software in terms of presenting information about azeotropes with identical qualitative composition (biazeotropy) were limited. The program allowed displaying the internal biazeotropy in binary [31,32] (ternary [30]) systems only in graphical form. There was no such possibility for quaternary systems. To predict the presence of internal singular points in such systems, it is necessary to involve qualitative methods of analyzing the internal structure. The most effective is the method given in [11,12] and based on the solution of the azeotropy rule, for example, in the form (2).

The results of modeling the phase equilibrium of the system considered with the parameters of the NRTL equation indicate the presence of one internal saddle-type azeotrope (Table 1). Thermodynamic-topological analysis of the diagram structure (determination of types and Poincaré indices of singular points) (Table 3) showed that a second internal azeotrope with the index “−1” should be present in the system, since the sum of the indices of singular points was +2, and it should equal zero for a quaternary system. Analysis of the structures of separatric manifolds (colored in green, blue and gray in Figure 3a) showed that it could only be a point of the “unstable node” type.

Thus, the phase diagram of the system corresponded to the structure (a) of Figure 3. These results were also confirmed by the calculation of phase equilibrium using the UNIFAC model (Appendix A). Different results from the presented ones were obtained for the UNIF-LL and NRTL-HOC models (Appendix A). In the first case, only one internal saddle-shaped azeotrope is modeled in the system (Figure 3b, Appendix A), in the second, no internal azeotrope was found (Figure 3c, Appendix A).

Despite the fact that all models adequately reproduced the boundary azeotropy, the question of the true internal structure of the phase diagram remained open. To solve it, a full-scale experiment was conducted to determine the presence/absence of one or two quaternary azeotropes. The key problem was the choice of the initial and conjugate compositions to determine the features of the boiling point change inside the composition tetrahedron. Analysis of the structure of the phase diagram showed that it was important to determine the type of ternary azeotrope ACN + CHEN + W in a tetrahedron (unstable node or saddle) and the temperature changes along the secant connecting this azeotrope to the point of pure CHL. At the same time, the secant was located near the line of the intersection of three two-dimensional separatric manifolds (highlighted in gray, green and blue in Figure 3). Depending on the nature of the boiling point change, one of the tetrahedron structures under consideration can be confirmed: the presence of two extremes (minimum and maximum) along the secant, Figure 3a; the presence of one extreme (maximum), Figure 3b; and the absence of extremes (descending curve), Figure 3c. Thus, the composition corresponding to the ternary azeotrope ACN + CHEN + W (Table 1) was selected as the initial one; titration was carried out with chloroform, the boiling point of which is lower than the temperatures in the ternary constituent ACN + CHEN + W.

The results of the temperature measurement are shown in the Table 4.

The analysis of the obtained dependence (Figure 4, blue dots) shows the presence of two extremes: a minimum of 334.24 K at a concentration of CHL 0.0534 mass frac. and a maximum of 335.95 K at 0.2143 mass frac. This fact indicates the presence of two internal azeotropes. Table 4 and Figure 4 (orange dots) also show the calculated boiling points for the mixture compositions obtained by adding different amounts of chloroform. The relative standard uncertainty in determining the calculated boiling point does not exceed 0.0035, which indicates the adequacy of mathematical modeling with the NRTL model.

## 4. Discussion

The results of this work show that there are certain problems in the study of systems with complex phase behavior using computer methods. For example, an adequate reproduction of the entire boundary azeotropy and phase equilibrium in the constituents of a multicomponent system cannot guarantee a correct description of the internal phase space, which was illustrated by the example of the system of acetonitrile + cyclohexene + water + chloroform. The capabilities of the software were limited in terms of the study of biazeotropic systems, especially for systems with more than three components, which required the use of qualitative methods (in particular, thermodynamic-topological analysis) to establish the structure of the vapor–liquid equilibrium diagram.

At the same time, combining the results of thermodynamic modeling and qualitative methods of phase diagrams studying made it possible to competently plan and minimize the amount of experimental work aimed at establishing the internal structure. 

In the present paper, a quaternary biazeotropy (two internal azeotropes in the phase diagram of the acetonitrile + cyclohexene + water + chloroform system) was discovered and experimentally confirmed for the first time. The type and boiling point of both quaternary azeotropes were determined: a saddle with 335.95 K, and an unstable node with 334.24 K. The maximum component of the internal azeotropes described earlier in the literature was equal to three (experimentally proven) [30]. Systems with a large number of components containing internal biazeotropy were not previously known.

Biazeotropic systems are distinguished by richer phase behavior compared to mono-azeotropic ones. Studies of such systems make it possible to understand the features of the diagrams’ structure, and possible ways of their evolution (for example, with pressure changes).

## Figures and Tables

**Figure 1 entropy-25-00980-f001:**
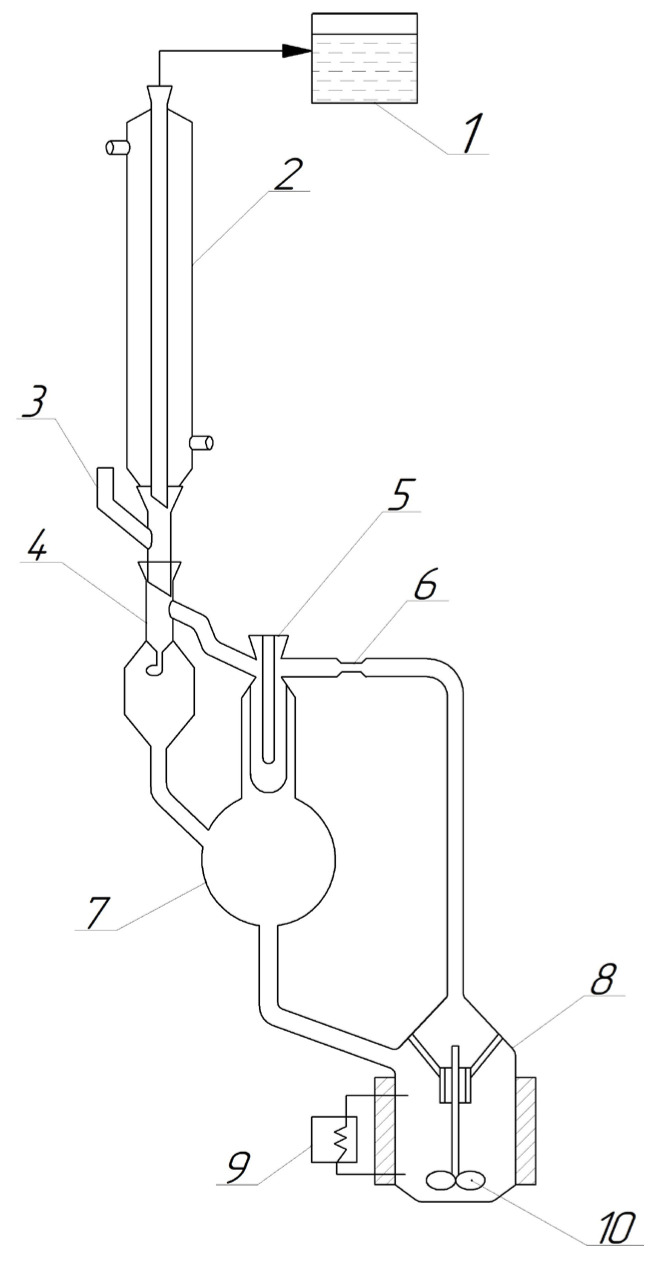
A modified Sventoslavskiy still (1—manostat; 2—condenser; 3—sample input device; 4—drop counter; 5—thermometer sleeve; 6—Cottrel pump; 7—separation space; 8—bottom; 9—electric heating; 10—magnetic stirrer).

**Figure 2 entropy-25-00980-f002:**
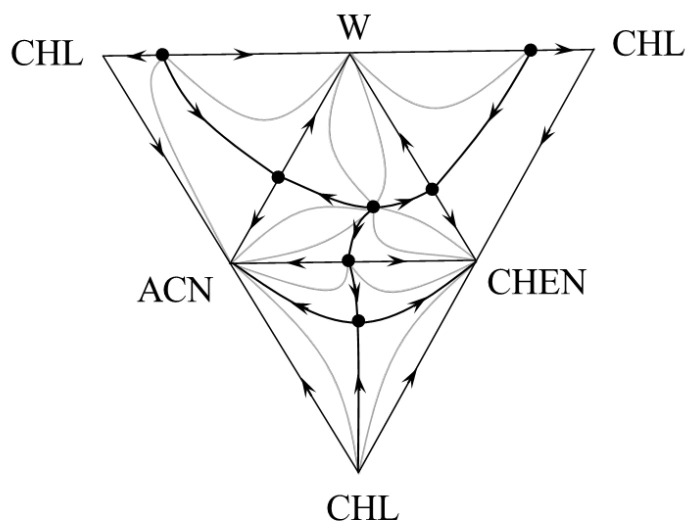
The structure of the scan of the phase diagram of the acetonitrile (ACN) + cyclohexene (CHEN) + water (W) + chloroform (CHL) system at 101.3 kPa.

**Figure 3 entropy-25-00980-f003:**
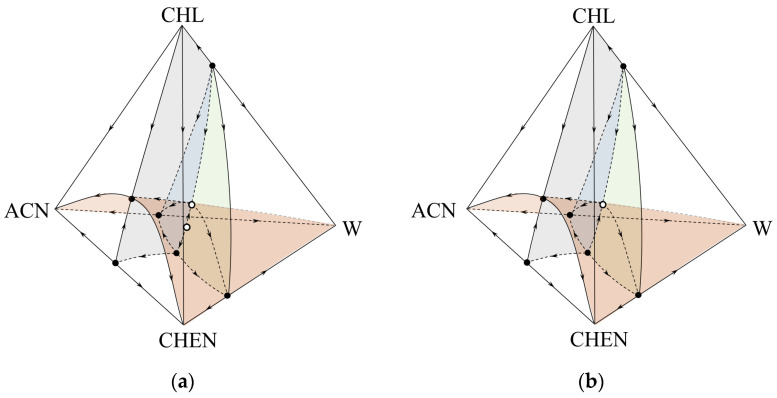
The possible structures of the phase diagram of the acetonitrile (ACN) + cyclohexene (CHEN) + water (W) + chloroform (CHL) system at 101.3 kPa: (**a**) NRTL, UNIFAC; (**b**) UNIF-LL; (**c**) NRTL-HOC.

**Figure 4 entropy-25-00980-f004:**
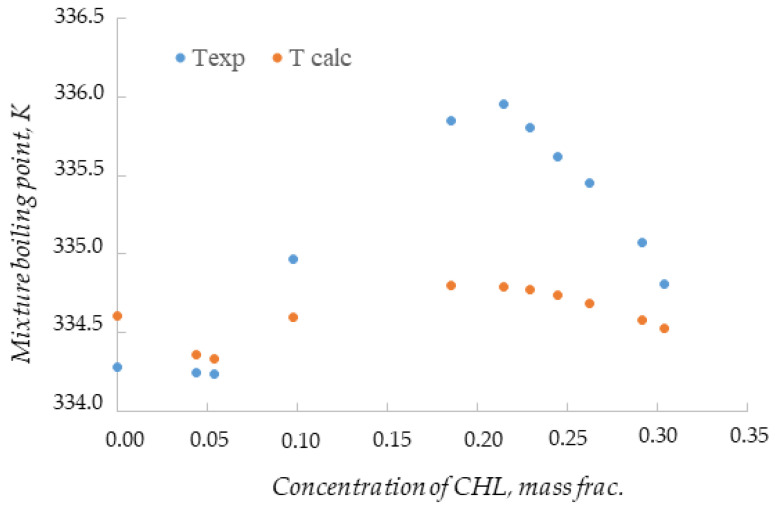
Dependence of the boiling point of the mixture on the concentration of chloroform.

**Table 1 entropy-25-00980-t001:** The comparison of experimental [1,19,21,24,25] and calculated (in Aspen Plus V.11.0) component boiling points and azeotropic data (composition, boiling point) of the acetonitrile (ACN) + cyclohexene (CHEN) + water (W) + chloroform (CHL) system at 101.3 kPa.

Component/Azeotrope	Experimental Data		Calculated Data		The Relative Standard Uncertainty	
	X_1_ (X_2_), Mole Frac.	T, K	X_1_ (X_2_/X_3_), Mole Frac.	T, K	for X_1_ (X_2_)	for T
ACN	1.0000	354.75	1.0000	354.80	-	0.0001
CHEN	1.0000	356.15	1.0000	356.03	-	0.0003
W	1.0000	373.15	1.0000	373.15	-	0.000
CHL	1.0000	334.30	1.0000	334.25	-	0.0001
ACN + W	0.6810	349.35	0.6735	350.06	0.011	0.002
ACN + CHEN	-	339.55	0.4874	338.85	-	0.002
CHEN + W	0.6910	343.95	0.6822	343.90	0.013	0.0001
CHL + W	0.8397	329.23	0.8360	329.29	0.004	0.0002
ACN + W + CHEN	-	334.35	0.3461 (0.1960)	334.61	-	0.0008
ACN + CHEN + CHL	-	-	0.3669 (0.3721)	340.23	-	-
ACN + W + CHEN + CHL	-	-	0.2901 (0.1968/0.3571)	334.61	-	-

**Table 2 entropy-25-00980-t002:** Properties of chemicals used in experiment at pressure (P) = 101 kPa.

Chemicals (CAS Number)	Source	Purity, Mass Frac.	Density *ρ*, g·cm^−3^		T, K	
			Exp.	Lit. [28]	Exp.	Lit. [28]
ACN (75-05-8)	Merck	≥0.9900	0.7862	0.7860	354.61	354.15
CHEN (110-83-8)	Merck	≥0.9900	0.8114	0.8110	326.20	356.15
W (7732-18-5)	-	1.0000	0.9990	1.0000	373.12	373.15
CHL (67-66-3)	Merck	>0.9900	1.4785	1.4800	334.39	334.65

T, temperature; *ρ*, density. The uncertainties, *u*, are *u*(T) = 0.05 K, *u*(P) = 0.1 kPa, and *u*(*ρ*) = 0.003 g/cm^3^.

**Table 3 entropy-25-00980-t003:** The types and Poincaré indexes of singular points of the phase diagram of the acetonitrile (ACN) + cyclohexene (CHEN) + water (W) + chloroform (CHL) system.

Singular Point	ACN	CHEN	W	CHL	CHL + W	ACN + CHEN	ACN + W
Type	N^st^	N^st^	N^st^	S	N^unst^	S	S
Poincaré index	+1	+1	+1	0	–1	0	–1
Singular Point	CHEN + W	ACN + CHL + CHEN		ACN + CHEN + W		ACN + CHEN + W + CHL	
Type	S	S		S		S	
Poincaré index	–1	–1		+1		+1	
The sum of Poincaré indexes is +2							

N^st^—stable node; N^unst^—unstable node; S—saddle.

**Table 4 entropy-25-00980-t004:** The results of experimental study of boiling point changes (mass of components in the initial mixture, g: CHEN—33.2470; CAN—12.5647; W—3.1146).

Mass CHL, g	Mixture Mass, g	Mixture Composition, Mass Frac.				T^exp^, K	T^calc^, K	u_r_
		CHEN	ACN	W	CHL			
0.0000	48.9263	0.6795	0.2568	0.0637	0.0000	334.28	334.61	0.0010
2.2357	51.1620	0.6498	0.2456	0.0609	0.0437	334.25	334.36	0.0003
2.7596	51.6859	0.6433	0.2431	0.0603	0.0534	334.24	334.33	0.0003
5.2943	54.2206	0.6132	0.2317	0.0574	0.0976	334.97	334.60	0.0011
11.1418	60.0681	0.5535	0.2092	0.0519	0.1855	335.85	334.80	0.0031
13.3445	62.2708	0.5339	0.2018	0.0500	0.2143	335.95	334.79	0.0035
14.5139	63.4402	0.5241	0.1981	0.0491	0.2288	335.80	334.77	0.0031
15.7888	64.7151	0.5137	0.1942	0.0481	0.2440	335.62	334.74	0.0026
17.4032	66.3295	0.5012	0.1894	0.0470	0.2624	335.45	334.69	0.0023
20.0747	69.0010	0.4818	0.1821	0.0451	0.2909	335.07	334.58	0.0015
21.3615	70.2878	0.4730	0.1788	0.0443	0.3039	334.81	334.53	0.0008

## Data Availability

Not applicable.

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
