# Peer review of "Experimental Evidence for Double Quaternary Azeotropy’s Existence"

_entropy, 2023, doi:10.3390/e25070980_

Round 1
Reviewer 1 Report
The article is devoted to the study of the structure of the phase equilibrium diagram of the acetonitrile - cyclohexene - water - chloroform system. The undoubted achievement of the authors is the discovery of two quaternary azeotropes. Previously such azeotropes with the number of components 2 and 3 were described in the literature, and ternary biazeotropy was experimentally confirmed relatively recently, in 2020 year. It should be noted that the authors began their research with a computational experiment and theoretical analysis, which allowed them correctly planning a full-scale experiment. The results obtained are beyond doubt. The research methods were chosen correctly and described quite fully, with the exception of the section ebulliometric titration. This section should be expanded somewhat.
The system example given in the article shows that when studying the phase equilibrium of multicomponent systems using computer methods, one cannot limit oneself to checking the adequacy of reproducing physicochemical properties in the boundary constituents of the system. It is clearly illustrated to what erroneous results this can lead. At the same time, the study of vapor-liquid equilibrium within the entire diagram is not justified and time-consuming. Is it possible to give practical recommendations on the organization of a full-scale experiment on the basis of the conducted research, or can this not be done only on the basis of studying one system?
Table 1 apparently contains the data obtained as a result of the calculation in AspenPlus. If this is the case, then it should be indicated in the name of the table, then it will be clear to the reader that the program produces only one of the internal azeotropes, and the authors found the second one based on theoretical analysis.
The authors write: «Analysis of the structures of separatric manifolds shows that it can only be a point of the "unstable node" type». But the analysis itself is not given in the text.
In general, the comments should be considered as minor revision.
Author Response
Dear Reviewer,
We express our gratitude to you for careful reading of our paper.
We checked the manuscript, followed your recommendations and added necessary explanations.
Here are our comments:
1) The research methods were chosen correctly and described quite fully, with the exception of the section ebulliometric titration. This section should be expanded somewhat.
The experimental section was expanded (Pages 4-5)
2) The system example given in the article shows that when studying the phase equilibrium of multicomponent systems using computer methods, one cannot limit oneself to checking the adequacy of reproducing physicochemical properties in the boundary constituents of the system. It is clearly illustrated to what erroneous results this can lead. At the same time, the study of vapor-liquid equilibrium within the entire diagram is not justified and time-consuming. Is it possible to give practical recommendations on the organization of a full-scale experiment on the basis of the conducted research, or can this not be done only on the basis of studying one system?
In general, we have previously published such recommendations in our papers (Reference 3 and 4) that describe in detail the procedure for the transition from the boundary space to the full phase diagram, in particular, depending on the structure of the tetrahedron scan (the presence of one or several ternary azeotropes), recommendations on the choice of conjugate composition for determining the type of ternary azeotropes (qualitative and quantitative composition) are formulated. For this reason, we do not duplicate this information in the present paper, but only illustrate the recommendations formulated earlier
3) Table 1 apparently contains the data obtained as a result of the calculation in AspenPlus. If this is the case, then it should be indicated in the name of the table, then it will be clear to the reader that the program produces only one of the internal azeotropes, and the authors found the second one based on theoretical analysis.
Of course, you are right, we have added this clarification (Page 2).
4) The authors write: «Analysis of the structures of separatric manifolds shows that it can only be a point of the "unstable node" type». But the analysis itself is not given in the text.
We have added relevant explanations in the text of the article. Clarifications have been added to the theoretical part (section 2.2) (Page 3)
With respect,
Authors of the paper
Reviewer 2 Report
This manuscript “Experimental evidence for double quaternary azeotropy existence” studied the phase equilibrium of acetonitrile + water + cyclohexene + chloroform system at 101.3kPa. Thermodynamic modeling in Aspen Plus V.10 was used to predict the internal structure of the constituent tetrahedron, and the existence of two internal quaternary azeotropes was confirmed by the results of full-scale experiments It can be published in “Entropy” after major revision. The concerns which should be considered by the authors are as follows:
1. In this manuscript, physical property methods such as NRTL, UNIFAC, UNIF-LL and NRTL-HOC are used respectively to calculate the azeotrope data of the four-element system. Please explain the differences and application scope of the four methods.
2. In the column of experimental data in Table 1, the azeotrope composition of acetonitrile and cyclohexene is not given, please give the reason.
3. In part 2.3.2, the description of the experimental process is very detailed, but it is suggested to add the schematic diagram of the experimental device to complement the description, so as to facilitate readers' understanding.
4. Azeotrope rules and vapor liquid equilibrium are important contents of the article. It is suggested to enrich the contents of the manuscript through the following references: Zhu et al., by studying liquid-liquid equilibrium data for four systems of n-heptane + acetone + 1-butyl-3-methylimidazole trifluoromethane sulfonic acid ([BMIM] [OTF]), n-heptane + acetone + 1-hexyl-3-methylimidazole trifluoromethane sulfonic acid ([HMIM]), Heptane + acetone + 1-butyl-3-methylimidazolium bis (((trifluoromethyl) sulfonyl) imide ([BMIM] [NTf2]) and heptane + acetone + 1-butyl-3-methylimidazolium were tested at T = 298.15K and P for hexafluorophosphate ([BMIM] [PF6]) = 101.325 kPa. By comparing the values of partitioning coefficient and separation factor, the experimental results show that the extraction ability of ILs as extraction solvent is in the order of [BMIM] [OTF]> [BMIM] [PF6 ]> [BMIM] [NTf2] and [BMIM] [OTF]> [HMIM] [OTF]. Finally, the NRTL and UNIQUAC models were used for regression to obtain the bivariate interaction parameters. (Journal of Chemical & Engineering Data, 2019, 64(3), 1202-1208); (Journal of Chemical & Engineering Data, 2019, 64(9), 4142-4147).
5. Authors should introduce the bebulliometric titration and explain why this method was chosen.
6. The introduction of this manuscript is too little to reflect the significance of the authors' completion of this work.
7. In Results, I have questions about how to construct the scans and complete structures that make up tetrahedrons, please explain them in detail
8. I noticed that the number of experiments in the manuscript is too small, and the obtained data may have errors. The authors should conduct several experiments to get accurate data. Zhuchkov et al. experimentally determined the existence of two ternary azeotropes in the ternary system(Journal of Chemical & Engineering Data, 2020, 65(4), 2002-2007); Upasana et al. used Antoine to model pure species data and compare them with experimental and literature values(Chemical Data Collections, 2023, 44, 101011); Pavlíček et al. used ebulliometric and static methods to process data based on newly measured total pressure of components(The Journal of Chemical Thermodynamics, 2020, 140, 105901).
10. The number of references cited by the authors is so small that the following references can be considered in order to increase the scientific nature of the manuscript:Aucejo et al. measured vapor-liquid equilibria in benzene (1) + hexafluorobenzene (2) systems(Journal of Chemical & Engineering Data, 1996, 41(1), 21-24); Dadmohammadi et al. evaluated the efficacy of the NRTL model to correctly characterize the conditions and properties of the LLE phase, including phase stability, miscibility, and fixed-point coordinates(Industrial & Engineering Chemistry Research, 2018, 57(21), 7282-7290); Wang et al. developed a comprehensive thermodynamic model for nitric acid-sulfuric acid-water ternary system based on previously published thermodynamic models of nitric acid-water and sulfuric acid-water binary systems with eNRTL equation(AIChE Journal, 2017, 63(7), 3110-3117).

It is suggested to revise the grammar of the article
Author Response
Dear Reviewer,
We are grateful for all the comments made, most of which we followed.
The explanations on each comment are given below.
- In this manuscript, physical property methods such as NRTL, UNIFAC, UNIF-LL and NRTL-HOC are used respectively to calculate the azeotrope data of the four-element system. Please explain the differences and application scope of the four methods.
Initially, we preferred the NRTL model, which has proven itself well when describing mixtures with strong deviations from ideal behavior. All binary and some ternary constituents of the system have been studied experimentally (including by us). Therefore, problems with the description of boundary properties were not expected (which was confirmed by a computer experiment). However, when we revealed (as a result of calculation and theoretical analysis) the presence of two internal azeotropes, the first thing that came to mind was that the model was inadequate (there are very few biazeotropic systems, only one ternary azeotropy has been experimentally confirmed). We tried to consider other models: UNIFAC (a group model, it is often used as a first approximation and moreover, in some cases it gives good results), UNIF-LL (has proven itself well in terms of describing systems with significant deviations of the liquid phase from the ideal behavior (our mixture contains not only azeotropes, but also areas of two-phase and three-phase splitting)), NRTL-HOC (the probability of non-ideal behavior of the vapor phase was assumed, for which the equation of state HOC was used). Despite the fact that some models have their limitations (for example, the UNIF-LL model - over a temperature range), we left them all for consideration, as long as with an adequate description of the boundary azeotropy, the models gave different results in the presence/absence of internal azeotropes. This is what prompted us to conduct a full-scale experiment in order to establish the true structure.
- In the column of experimental data in Table 1, the azeotrope composition of acetonitrile and cyclohexene is not given, please give the reason.
Experimental data on the composition of this azeotrope are not available in the literature, only the boiling point. We did not set the task of determine the composition. However, our calculated composition of this azeotrope coincides with the results of calculations by other authors and the results of modeling the system phase equilibrium using the UNIFAC model.
- In part 2.3.2, the description of the experimental process is very detailed, but it is suggested to add the schematic diagram of the experimental device to complement the description, so as to facilitate readers' understanding.
This part has been expanded a little more (recommendations of other reviewers). We have added a figure of the devise used (Figure 1).
- Azeotrope rules and vapor liquid equilibrium are important contents of the article. It is suggested to enrich the contents of the manuscript through the following references: Zhu et al., by studying liquid-liquid equilibrium data for four systems of n-heptane + acetone + 1-butyl-3-methylimidazole trifluoromethane sulfonic acid ([BMIM] [OTF]), n-heptane + acetone + 1-hexyl-3-methylimidazole trifluoromethane sulfonic acid ([HMIM]), Heptane + acetone + 1-butyl-3-methylimidazolium bis (((trifluoromethyl) sulfonyl) imide ([BMIM] [NTf2]) and heptane + acetone + 1-butyl-3-methylimidazolium were tested at T = 298.15K and P for hexafluorophosphate ([BMIM] [PF6]) = 101.325 kPa. By comparing the values of partitioning coefficient and separation factor, the experimental results show that the extraction ability of ILs as extraction solvent is in the order of [BMIM] [OTF]> [BMIM] [PF6 ]> [BMIM] [NTf2] and [BMIM] [OTF]> [HMIM] [OTF]. Finally, the NRTL and UNIQUAC models were used for regression to obtain the bivariate interaction parameters. (Journal of Chemical & Engineering Data, 2019, 64(3), 1202-1208); (Journal of Chemical & Engineering Data, 2019, 64(9), 4142-4147).
One of the references have been added (ref. 18)
- Authors should introduce the ebulliometric titration and explain why this method was chosen.
The choice of this method is primarily due to the device design feature, in particular related to the possibility of intensive mixing of the liquid (the system contains areas with two and three liquid phases). It is much more difficult to carry out such mixing in a column (distillation analysis).
We have added the explanation in the text (Page 4)
- The introduction of this manuscript is too little to reflect the significance of the authors' completion of this work.
The introduction has been somewhat expanded by us, the corresponding references to the literature have been added.
- In Results, I have questions about how to construct the scans and complete structures that make up tetrahedrons, please explain them in detail.
Our scientific group (led by prof. L.A. Serafimov) has been developing methods for studying the phase space of multicomponent mixtures for a long time. Despite the fact that the fundamental foundations of this theory were laid back in the 1960s and 1970s, we managed to make some progress in this direction. In particular, we have developed a methodology for studying the VLE diagrams, which is based on the analysis of the boundary space, the prediction of possible structures of the internal space of the diagram. It is based on determining the types and indices of singular points located on scans of different dimensions (for an n-component system, an n-2 dimension scan is important). The determination of the fact of the presence of an internal singular point is carried out by solving the azeotropy rule (as indicated in this article). Both the scan itself and the diagram (after determining the presence / absence of an internal azeotrope) are constructed according to data on boiling points (general rules for constructing distillation curve diagrams). In this article, we did not discuss the methodology in detail, since it had already been published by us earlier, only references are given in the article.
- I noticed that the number of experiments in the manuscript is too small, and the obtained data may have errors. The authors should conduct several experiments to get accurate data. Zhuchkov et al. experimentally determined the existence of two ternary azeotropes in the ternary system(Journal of Chemical & Engineering Data, 2020, 65(4), 2002-2007); Upasana et al. used Antoine to model pure species data and compare them with experimental and literature values(Chemical Data Collections, 2023, 44, 101011); Pavlíček et al. used ebulliometric and static methods to process data based on newly measured total pressure of components(The Journal of Chemical Thermodynamics, 2020, 140, 105901).
We have expanded the description of the experimental part, indicating that the full-scale experiment was carried out using two identical devices. The results obtained on both devices fall on the same curve (in both cases, the minimum and maximum are fixed).
We previously used these devices to determine the presence of two ternary azeotropes (including at different pressures) in the benzene-perfluorobenzene-water system. These results at 101.3 kPa were published in 2020 in the journal JChemEngData and at other pressures are planned to be published in 2023 (the article has been accepted for publication and will be published in the near future).
- The number of references cited by the authors is so small that the following references can be considered in order to increase the scientific nature of the manuscript: Aucejo et al. measured vapor-liquid equilibria in benzene (1) + hexafluorobenzene (2) systems(Journal of Chemical & Engineering Data, 1996, 41(1), 21-24); Dadmohammadi et al. evaluated the efficacy of the NRTL model to correctly characterize the conditions and properties of the LLE phase, including phase stability, miscibility, and fixed-point coordinates(Industrial & Engineering Chemistry Research, 2018, 57(21), 7282-7290); Wang et al. developed a comprehensive thermodynamic model for nitric acid-sulfuric acid-water ternary system based on previously published thermodynamic models of nitric acid-water and sulfuric acid-water binary systems with eNRTL equation(AIChE Journal, 2017, 63(7), 3110-3117).
In our article in JChemEngData (2020) an overview of experimental data on phase equilibrium in biazeotropic systems was given, problems of modeling such systems were discussed. For this reason, in order not to duplicate the already published material, we have limited ourselves to those references that are given in the article.
One of the above references has been added because it is relevant to this article (regarding the NRTLmodel).
Besides we have expanded the list of references by including articles discussing the thermodynamic modeling problems.
With respect,
Authors of the paper
Reviewer 3 Report
see attached

just minor concern outlined on attachement.
Author Response
Dear Reviewer,
We are grateful for the high appreciation of our research and for valuable comments about our paper. We have added necessary explanations.
1) The phenomena of retrograde condensation (evaporation) are observed locally near critical points under appropriate conditions. The values of the pressure and boiling point of the mixtures during our full-scale experiment are significantly lower than the critical ones. In addition, stable maintenance of the equilibrium temperature for at least 30 minutes after each addition of chloroform was noted, which, in our opinion, indicates the absence of non-trivial effects of vapor-liquid equilibrium.
2) We left the dots of the internal azeotropes unfilled in Figure 2, since the main question is precisely their existence.
We have added the description of experimental part: The task of ebulliometric titration was to add portion-small amounts of chloroform to the ternary mixture of ACN + CHEN + W (to its total concentration of about 30% mass.) and determine the trend of boiling point changes. The experiments were carried out in parallel in two identical devices; the time for equilibrium achieving, during which the parameters did not change, was at least 30 minutes. The trend of the boiling point change was almost the same; the points fall on the same curve. The latter is also confirmed by mathematical modeling (Pages 4-5)
The boiling point remained constant until the addition of the next portion of chloroform during the experiments. We recorded a gradual decrease and then an increase in the boiling point. The concentration of chloroform at the point of the minimum boiling point exceeds 5% mass., i.e. subsequent compositions actually "go" inside the tetrahedron. And we would have to record an increase in the boiling point in the absence of a second internal azeotrope, which did not happen.
3) The peculiarity of the distillation analysis is associated with the presence of two- and three-phase splitting areas in the system, which makes the contact of phases in the column less effective: different layers wet the nozzle differently, it is more difficult to organize intensive mixing of the splitting mixture in the column than in the ebulliometer. The latter reduces the accuracy of the results obtained. For these reasons, preference is given to ebulliometric titration. A closer contact of the liquid phases is ensured (the liquid becomes more homogeneous) in an ebulliometer using a magnetic stirrer, i.e. the regime of intensive mixing of the splitting mixture is implemented, the liquid-vapor phase transition is more stable and the composition of the equilibrium vapor (azeotrope composition), in our opinion, is determined more accurately.
4) Thank you, we checked the text.
With respect,
Authors of the paper

Round 2
Reviewer 1 Report
The authors have made the necessary changes to the text of the article. The article can be accepted for publication in this form.
Reviewer 2 Report
All comments have been revised. I suggest that it be published on Entropy
Reviewer 3 Report
I will agree to disagree with authors particularly with attention to the last comment in their author response. If stirring is needed to keep the liquid 'more' homogeneous, then how are the azeotropes homogeneous. If I shake a bottle of oil and water, it can appear homogeneous. If I let the bottle stand for a few minutes without shaking, there is phase separation. The authors should just state what type of internal azeotropes their paper is addressing - homogeneous or heterogeneous - just so it is perfectly clear to the readers.